# Apoe4 and Alzheimer’s Disease Pathogenesis—Mitochondrial Deregulation and Targeted Therapeutic Strategies

**DOI:** 10.3390/ijms24010778

**Published:** 2023-01-01

**Authors:** Mariana Pires, Ana Cristina Rego

**Affiliations:** 1Center for Neuroscience and Cell Biology (CNC), University of Coimbra, Polo I, 3004-504 Coimbra, Portugal; 2Faculty of Medicine, University of Coimbra, Polo III, 3004-354 Coimbra, Portugal

**Keywords:** mitochondrial function, mitochondrial morphology, gene expression, dementia, aging, neuroinflammation, transcription

## Abstract

*APOE* ε4 allele (ApoE4) is the primary genetic risk factor for sporadic Alzheimer’s disease (AD), expressed in 40–65% of all AD patients. ApoE4 has been associated to many pathological processes possibly linked to cognitive impairment, such as amyloid-β (Aβ) and tau pathologies. However, the exact mechanism underlying ApoE4 impact on AD progression is unclear, while no effective therapies are available for this highly debilitating neurodegenerative disorder. This review describes the current knowledge of ApoE4 interaction with mitochondria, causing mitochondrial dysfunction and neurotoxicity, associated with increased mitochondrial Ca^2+^ and reactive oxygen species (ROS) levels, and it effects on mitochondrial dynamics, namely fusion and fission, and mitophagy. Moreover, ApoE4 translocates to the nucleus, regulating the expression of genes involved in aging, Aβ production, inflammation and apoptosis, potentially linked to AD pathogenesis. Thus, novel therapeutical targets can be envisaged to counteract the effects induced by ApoE4 in AD brain.

## 1. Introduction

Alzheimer’s disease (AD) is a progressive neurodegenerative disease responsible for 60–80% of all cases of age-dependent dementia. AD affects about 50 million people worldwide and its prevalence is expected to increase by almost 70% by 2050 [1,2,3]. Considering that there is currently no protective therapeutic strategy (ies) to improve the clinical symptoms and extend patients’ survival [4], these numbers reinforce the importance of studying its mechanisms and urgently identify therapeutical targets.

AD is characterized by progressive preclinical stage, completely asymptomatic to a state of mild cognitive impairment (MCI), characterized by a slight cognitive decline, which evolves to a stage of dementia. This cognitive decline is associated with progressive memory loss, which is the most typical symptom associated with the disease, changes in executive function, which affects the ability to perform routine daily activities, and impairments in awareness, language, social behavior and visuospatial function [2,5,6]. AD is considered one of the main causes of disability in the elderly population [2,5]. Indeed, aging is the strongest risk factor for the sporadic or late-onset AD (LOAD), the most common form of the disease, affecting nearly 95% of all patients. In LOAD the symptoms usually arise after 65 years of age. Although the main risk factor for LOAD is aging, there are also some environmental factors, namely microbial infections, dietary and sleep habits and psychological stress, and also genetic variations. Among these genetic polymorphisms is ApoE4, which is one of the three alleles for apolipoprotein E (*APOE*) gene—ApoE2, ApoE3 and ApoE4 [2,5,7].

Familial or early-onset AD (EOAD) is relatively rare, affecting 1 to 5% of all patients and it usually arises before 65 years of age. Although the majority of EOAD cases are linked to unidentified genetic variations, 10–15% of them are caused by known mutations in amyloid precursor protein (*APP*), presenilin 1 (*PSEN1*) or presenilin 2 (*PSEN2*) genes, that are inherited in an autosomal dominant manner. Among those, more than three hundred mutations have been identified, causing, respectively, 10 to 15% (*APP*), 30 to 70% (*PSEN1*) and less than 5% (*PSEN2*) of the autosomal dominant EOAD cases [8].

When considering cellular and molecular alterations, AD starts 25–30 years before symptoms arise [5]. The main hallmarks of the disease are extracellular senile plaques, composed by the amyloid-β peptide (Aβ) and intraneuronal neurofibrillary tangles (NFTs), composed by hyperphosphorylated tau protein (P-tau). Apart from these protein aggregates, AD is also characterized by synaptic disruption and mitochondrial dysfunction, which hypothetically underly neuronal death and brain atrophy [1,2]. Longitudinal studies in EOAD patients have helped to understand the sequence of AD molecular alterations over time. It has been suggested that Aβ aggregation is the first change to occur, starting nearly 25–30 years before symptomatic onset, followed by neuronal metabolic disruption around 17 years before symptoms arise; brain atrophy, particularly at the hippocampal region, was suggested to occur about five years before the symptoms, with P-tau accumulation in NFTs being the molecular alteration that is closest to the symptoms onset [9].

Memory impairment and cognitive decline in AD are strictly related to region-specific alterations, namely the presence of protein aggregates, mostly in memory-related brain areas and brain atrophy. In particular, the hippocampal region, entorhinal cortex and neocortex form a neuronal network that is progressively affected in AD [9]. Global brain atrophy is caused by neurodegeneration, loss of neuronal processes, decreased number of synapses and ultimately neuronal death [10]. The earliest molecular alterations in AD start at the hippocampal region, with the extracellular Aβ accumulation and the death of CA1 pyramidal neurons being the most evident manifestations [8,11]. Later, along AD progression and apart from the previous changes, neuronal death and the formation of NFTs in layers II and IV of entorhinal cortex are the most marked molecular alterations, typically accompanied by MCI symptoms [12].

According to the tau hypothesis, which postulates that tau aggregation is the main factor underlying AD development, tau protein propagates within the brain in a manner that is correlated to symptoms severity [13]. Based on this hypothesis, the most advanced and severe stage of AD, which is clinically characterized by a stage of dementia, is then linked to P-tau propagation from the entorhinal cortex to the neocortex, and consequent presence of NFTs in this brain region [2]. Apart from these alterations, Aβ aggregation and neuronal death also occur at the neocortex at this stage of the disease [5].

The activation of glial cells, associated to neuroinflammation, is also a relevant feature of AD. Glial cells, namely astrocytes and microglia, are crucial to maintain brain homeostasis. Astrocytes support neuronal metabolism and, together with microglia, play an important immune function. Under cellular stress conditions, namely in AD, these cells become activated, assuming a pathological role. Among other impairments, active astrocytes lose the capacity to support neuronal energetic demand, and active microglia enhance the production of pro-inflammatory cytokines and lose the ability to phagocytize toxic species, namely Aβ. These changes contribute to neuroinflammation and consequently to neurodegeneration [14]. Recent studies have also suggested that P-tau inclusions may occur in hippocampal astrocytes, contributing to synaptic alterations, cognitive decline and AD progression [10].

As previously described in this section, ApoE4 has been associated to many pathological processes possibly linked to AD neurodegeneration. However, the exact intracellular and molecular mechanisms underlying ApoE4 neurotoxicity and pathogenic effects are not completely understood. Some emerging hypotheses have been raised in the context of mitochondrial-related impairments associated to ApoE4 expression. Thus, this review aims to detail the impact of this genetic risk factor for LOAD and mitochondrial dyshomeostasis, as well as ApoE4-targeted therapeutic strategies based on ApoE4 potential mechanisms of action.

## 2. Role of ApoE in AD Pathogenesis

*APOE* ε4 allele (ApoE4) is the strongest genetic risk factor for sporadic AD [15] and an important biomarker of susceptibility for this disease [16]. Even though the global frequency of ApoE4 is approximately 13.7%, around 40% of all AD patients carry at least one copy of the ε4 allele [2]. Carrying one copy of this allele increases the risk of developing AD by about 3 to 7 times, while carrying two copies increases the risk up to 12 times [17]. The *APOE* ε4 allele also reduces the age of onset of AD-associated cognitive impairment by 7 to 9 years per copy [18]. Factors such as family history, sex, serious head injury, smoking, cholesterol levels and estrogen may modify the ApoE-related risk [16], which may help to understand why not all carriers develop AD and why not all AD patients carry this allele. Being a risk factor, ApoE4 rather increases the susceptibility to AD than determines its development [19]. By contrast, carrying one copy of the *APOE* ε2 allele (ApoE2) reduces the risk of developing AD by about 40% [17], being a protective factor against its sporadic form. However, this is the least common allele, presenting a global prevalence of 0–20% [20]. *APOE* ε3 allele (ApoE3) is the most frequent one, being expressed by 60–90% of the world population [20], but has no significant effect on AD risk [18].

These three isoforms differ in simple substitutions of amino acids (aa) involving cysteine and arginine residues at positions 112 and 158 at the N-terminal. While ApoE3 has a cysteine at residue 112 and an arginine at residue 158, ApoE4 has an arginine residue and ApoE2 has a cysteine residue at both positions. These variations show up as changes in the tertiary structure of these 34 kDa (299 aa) proteins [21]. In particular, ApoE4 assumes a pathological conformation due to the interaction between arginine −61 and glutamate −255, at the N-terminal and the C-terminal domains, respectively [22]. Due to this domain interaction, ApoE4 is susceptible to proteolytical cleavage and the resulting fragments are suggested to play an essential role in ApoE4 neurotoxicity [23].

## 3. ApoE4 Proteolytic Cleavage

As previously described, ApoE4 shows an exclusive interaction between the N- and C-terminal domains, which makes this protein the least stable among the three ApoE isoforms [24]. Due to this structural property, ApoE4 is more susceptible to proteolytical cleavage than ApoE2 and ApoE3 [23]. It has been hypothesized that, under stress conditions, ApoE4 proteolytical cleavage is favored and the resulting products are thought to underlie AD pathogenesis and the risk of dementia [25].

Several proteases have been suggested to be responsible for ApoE4 cleavage at different sites, including collagenase, matrix metalloproteinas-9 (MMP-9) [25], neuron-specific α-chymotrypsin-like serine protease [26], cathepsin D (the only aspartic protease identified) [27] and high-temperature requirement serine protease A1 (HtrA1) [21]. Some of these enzymes are cytoplasmatic, while others can cleave ApoE4 extracellularly, generating the fragments that are further endocytosed, enter the cytosol and interact with mitochondria, intracellular proteins or translocate to the nucleus [11].

Fragments with 14 to 20 kDa have been found in the brain of AD patients carrying the *APOE* ε4 allele [27], including ApoE4 N-terminal fragment [25] and C-terminal truncated forms [28], and in higher levels than in age- and sex-matched controls [29]. Additionally, it has been observed that ApoE4 fragmentation occurs in a brain region-dependent way, preferentially in the neocortex and hippocampus of neuron-specific enolase (NSE)-ApoE4 transgenic mice (mice expressing ApoE4 under the control of NSE promoter), being both areas very susceptible to AD-related neurodegeneration [30], as previously described in this review.

Although, in general, ApoE4 fragments are proposed to promote neuronal dysfunction, neurodegeneration and astrogliosis [11], it has been suggested that certain fragments are more likely related to some pathogenic features of AD than others. For example, some studies in N2a cells demonstrated that ApoE4 N-terminal fragments (1–272 aa) interact with mitochondria and may possibly impact on mitochondrial function and cell viability [29], as well as Aβ deposition [21] and AD-related tau pathology [11]. Other authors demonstrated that N-terminal fragments of ApoE4 (1–151 aa) translocate to the nucleus and act as a transcription factor (Figure 1), promoting AD pathogenesis [25], but also being taken up by microglia and contributing to cell death [27].

Within the N-terminal domain, ApoE4 receptor-binding region (136–150 aa) is required for its binding to the low-density lipoprotein (LDL) receptor and to tau protein, while the lipid-binding region (240–270 aa) in the C-terminal domain is more likely involved in direct interaction with mitochondria (Figure 1) and binding to Aβ [26]. Both regions are thought to be critical for ApoE4 fragment-related neurotoxicity and mitochondrial impairment, rather than full-length protein [29,31]. Although previous studies have identified aa 273–299 as neuroprotective [29], more recently it has been demonstrated that these ApoE4 fragments are associated to changes in mitochondrial dynamics and morphology, endoplasmic reticulum (ER) stress, decreased ATP production and mitochondrial membrane potential (Δψ_m_) and increased ROS levels, in N2a cells [28]. The above mechanisms will be further elucidated in this review.

Based on these observations, contradictory ApoE4 data may be accounted for by the differential effects exerted by each one of its fragments.

## 4. Physiological and Pathological Roles of ApoE in the Central Nervous System

Apolipoproteins, namely ApoE, are the protein component of plasma lipoprotein complexes. These components play critical roles in preserving the structural integrity of lipoproteins, acting as ligands for lipoprotein receptors and facilitating their solubilization in the blood [20,32]. The complex formed by apolipoproteins and plasma lipoproteins acts as a carrier for lipids and cholesterol between cell types or tissues [33].

Up to 75% of ApoE synthesis occurs in the liver [20], but it is also significantly produced in the brain, where ApoE is the most abundant apolipoprotein [18]. In the central nervous system (CNS) and under physiological conditions, ApoE is mostly produced by astrocytes and microglia [2], mediating cholesterol transport to neurons via low density lipoprotein receptors (LDLR) [33].

Taking into consideration that cholesterol and other lipids are crucial for axonal growth, synaptic formation and plasticity, the specific interaction between ApoE and LDLR and the consequent control of circulating cholesterol levels are key mechanisms to support membrane homeostasis, synaptic integrity, damage repair, being crucial for learning and memory processes [32,33].

In this context, the efficacy of ApoE-mediated lipid transport in the brain is isoform-dependent. For example, previous studies have demonstrated that ApoE4-target replacement (TR) mice express lower levels of total ApoE than ApoE3-TR mice, mainly through a reduction in ApoE4 levels, which was proposed to contribute to AD progression through impaired neuronal and synaptic homeostasis [34]. ApoE4 was also suggested to be less efficient than ApoE3 in transporting brain cholesterol [33], which may underly the reduction in cholesterol levels in patients with AD, particularly in hippocampal and cortical areas [35] that are highly affected in this pathology.

The mechanisms underlying the association between ApoE4, lipid flux deregulation and AD-related cognitive decline remain largely unclear. A recent study demonstrated a link between ApoE4 and impaired neuronal myelination, underlying dysfunctional electrical activity in neurons and the subsequent impact on learning and memory processes [36].

As previously referred, ApoE is mainly produced by glial cells under physiological conditions. By contrast, under brain stress conditions, namely due to oxidative stress, aging and/or Aβ toxicity, astrocytic production becomes exacerbated and neurons are also stimulated to produce ApoE as a compensatory mechanism to restore brain homeostasis [2]. However, in the case of APOE ε4 carriers, neuronal expression of ApoE4 has a neurotoxic effect [22], accelerating neurodegeneration and contributing to AD progression.

Although it is known that ApoE4 is an important risk factor for AD, its mechanisms of action are not completely known yet, although several mechanisms have been proposed. Some authors postulate that ApoE4 is linked to the major molecular hallmarks of AD, namely Aβ aggregation and clearance, NFTs and tau-related neurodegeneration. Moreover, emerging hypotheses have also been linking ApoE4 to disturbed synaptic plasticity, glial response and neuroinflammation, blood-brain barrier (BBB) disruption [17] and astrocyte-associated lipid metabolism [2], while emphasizing the neurotoxic effects of ApoE4 fragments [25]. Furthermore, previous studies have linked each of these pathological alterations and brain aging to mitochondrial changes, which in turn links ApoE4 to alterations in this organelle [22] and to AD progression. Apart from these mechanisms, recent studies postulate that ApoE4 may act as a transcription factor, regulating the expression of genes potentially involved in cellular and molecular pathways related to AD development [37]. The effect of ApoE4 on mitochondrial function and dynamics and its effect on gene expression will be further explored in this review.

### 4.1. Aβ—ApoE4 Pathogenic Mechanisms

In general, ApoE4 pathogenic mechanisms can be divided into Aβ-related and Aβ-independent mechanisms. Primary studies showed that ApoE4 is associated with a greater accumulation of Aβ peptide, comparing to other ApoE genetic variations. Besides that, ApoE4 is linked to an increase in Aβ synthesis, promoting APP processing through mitogen-activated protein kinase (MAPK) signaling [24], Aβ seeding [2] and Aβ aggregation into oligomers and fibrils, a process involved in the so-called ‘amyloid-β cascade’ [22]. ApoE4 is able to directly interact with both soluble and fibrillar forms of Aβ [17,38], accelerating its oligomerization and deposition in the brain [2].

In humans, it was shown that carrying the ε4 allele of *APOE* gene is linked to an earlier and more extensive formation of senile plaques [25], associated to lower levels of Aβ_1–42_ in cerebrospinal fluid (CSF) [2]. It has also been observed that induced pluripotent stem cells (iPSC)-derived neurons from LOAD patients expressing ApoE4 secrete higher levels of Aβ, when compared to ApoE3-expressing neurons, which agrees with Aβ accumulation in the brain [39].

Apart from the direct interaction between ApoE4 and Aβ species, Aβ accumulation and increased levels of extracellular senile plaques may be justified by defective clearance and astrogliosis. As previously mentioned, astrocytes play a fundamental immune role in the CNS, namely through Aβ clearance.

As explained before, glial cells secrete ApoE to the extracellular space and ATP-binding cassette transporter A1 (ABCA1) mediates its lipidation, leading to the formation of a lipidated ApoE-Aβ complex. In other words, ABCA1 facilitates the interaction between ApoE and lipoproteins, cholesterol or phospholipids, which in turn enhance its interaction with soluble and fibrillar Aβ species [17,40]. Under physiological conditions, this complex binds to LDLR and LDL receptor-related protein-1 (LRP1) at the cell surface of glial cells and neurons, that further uptake Aβ and promote its clearance from the extracellular space. In the case of astrocytes, Aβ can be enzymatically degraded by neprilysin (NEP, an integral membrane protein residing in the plasma membrane, with the active site located in the extracellular space, thus considered an ectoenzyme) or extracellular matrix metalloproteinase-9 (MMP-9, involved in the degradation and remodelling of the extracellular matrix) [41], which prevents extracellular Aβ aggregation into oligomers and fibrils.

However, the ability of glial cells to phagocytize Aβ is dependent on the ApoE isoform. On the one hand, the exacerbated production of ApoE4 impairs astrocytic function, which disrupts its immune role in the CNS. For example, it has been proposed that carrying at least one copy of the *APOE* ε4 allele impairs the astroglial response to Aβ plaques, which is in turn correlated to cognitive decline [42].

On the other hand, ApoE4 competes for LDLR and LRP1 at the glial cells and neuronal surface, that are common to Aβ, preventing Aβ clearance and leading to its accumulation, that in turn enhance its oligomerization and the formation of senile plaques [43]. Besides that, ApoE4 also prevents Aβ enzymatic degradation, through the blockade of astrocytic NEP and MMP-9, or the extracellular insulin-degrading enzyme (IDE). Some authors have demonstrated that LOAD patients carrying the ApoE4 allele, as opposed to ApoE3 carriers, show a decrease by about 50% in the levels of hippocampal IDE protein, which may explain the Aβ accumulation in this brain area [43].

Thus, although recent studies in iPSC-derived cerebral organoids derived from LOAD patients show that Aβ clearance mechanism rather than APP processing is altered, independently on ApoE4 status [15], previous studies have shown that ApoE4 may interfere with Aβ metabolism, either by directly binding to the peptide or preventing its clearance, which promotes its deposition in toxic forms and potentiates AD development. These mechanisms may help to understand why ApoE4 carriers develop AD and cognitive impairment earlier in life.

### 4.2. ApoE4 and AD Development

Considering the ApoE4 mechanisms that are not directly dependent on Aβ, some studies have shown an association of ApoE4 per se with neurodegeneration, neurotoxicity, synaptic changes, aging and cognitive impairments [40].

As it was previously explained in this review, astrocytes are essential to support neuronal survival and subsequent synaptic integrity. Apart from the impaired capacity to phagocytize Aβ, astrocytes expressing ApoE4 have been linked to reduced neurotrophic support [17]. Specifically, human iPSC-derived astrocytes showed differential effects on supporting synaptogenesis and neuronal survival according to the *APOE* allele being expressed, being this support decreased in ApoE4-expressing astrocytes. Although these cells secrete specific synaptogenic and trophic molecules, such as brain-derived neurotrophic factor (BDNF), cholesterol-ApoE binding was pointed crucial in mediating the referred neurotrophic support [44]. Contrary to these observations, it has been observed that neuronal cultures derived from iPSC from LOAD patients expressing ApoE4, relatively to ApoE3, show increased number of synapses and augmented frequency of mini-excitatory postsynaptic current (mEPSC), which suggests a higher synaptic activity through neurotransmitter release, correlated with increased Aβ release [39]. Besides that, ApoE4 is less prone to lipidation by ABCA1, as previously mentioned, being in a hypolipidated state that is thought to impair lipid metabolism, dysregulate the levels of cerebral cholesterol and phospholipids and compromise synaptic integrity [45].

Indeed, synaptic integrity is crucial for learning and memory [40], and thus the mechanisms underlying ApoE4-mediated impaired neuronal activity and cognitive function have been the focus of investigation. As already mentioned in this review, a recent study has demonstrated that the impaired ability of ApoE4 to regulate cholesterol flux and metabolism is linked to decreased myelination, which may underly cognitive deficits. Specifically, the authors observed, in human post-mortem brain tissues, that ApoE4 is associated to cholesterol-related transcriptomic alterations and an aberrantly increased cholesterol accumulation in oligodendrocytes. Brain tissues from ApoE4 carriers showed decreased myelin levels [36], concordantly with previous studies demonstrating altered white matter patterns in AD brains [46]. ApoE4 TR-mice also showed an excessive cholesterol accumulation in oligodendrocytes and decreased axonal myelination [36]. Thus, ApoE4-mediated changes in cholesterol homeostasis and flux in oligodendrocytes disturb axonal myelination, which may be related to impaired electrical activity in neurons and learning and memory [36].

Previous studies have also demonstrated that astrocytes derived from ApoE4-KI (knock-in) mice display impaired synaptic pruning related to the reduced ability to phagocyte C1q, which is a protein involved in the CNS immune synaptic elimination. Thus, ApoE4-expressing astrocytes are associated to the accumulation of senescent synapses and accelerated neurodegeneration [17,47]. In the context of ApoE4-induced neurodegeneration, this ApoE isoform also induces excessive Ca^2+^ levels and neuronal death [38], as further discussed in this review, as well as increased markers of cellular stress. iPSC-derived neurons from LOAD patients showed increased levels of binding immunoglobulin protein (BiP) or GRP78, caspase-3 and -4, and ROS, as indicators of ER-stress, apoptosis and oxidative stress, respectively [15,48]. These neurodegenerative mechanisms may underlie the decrease in cortical thickness and hippocampal size that have been observed in ApoE4 carriers [2]. For example, early magnetic resonance imaging (MRI) studies in cognitively normal individuals expressing the *APOE* ε4 allele have revealed a decrease in hippocampal area, that was accompanied by a lower performance on neuropsychological memory tests, when compared to ApoE3 carriers [49].

Another ApoE4-mediated effect at the synaptic level, are the decreased dendritic spine density and dendrite length, inhibition of neurite outgrowth and impaired neuronal plasticity [43], particularly in hippocampal pyramidal neurons [38]. The loss of synaptic integrity has been also observed in cerebral organoids derived from AD patients, expressing ApoE4 [15]. Other studies have demonstrated a decrease in the number of GABAergic interneurons in dentate gyrus, potentially linked to ApoE4 neurotoxic fragments, leading to neuronal overexcitation and excitotoxicity [11].

Altogether, these alterations in neuronal function compromise normal network activity and trigger cognitive alterations. In previous studies, it was observed that NSE-ApoE4 transgenic mice showed deficits in learning and memory, in an age-dependent way, and these changes were more accentuated when ApoE4 was expressed by neurons rather than glial cells, when compared to NSE-ApoE3 mice [22]. Moreover, the ApoE4-related synaptic changes can be linked to spatial and work memory impairment [38].

### 4.3. Tau—ApoE4 Pathogenic Mechanisms

Regarding AD major molecular hallmarks, ApoE4 is not only related to Aβ, but also to tau protein. This protein is suggested to play a role in regulating microtubule assembly and consequent cargo transport along the neurons, and in maintaining axonal structural integrity. In AD, tau is exposed to aberrant phosphorylation by diverse kinases, namely glycogen synthase kinase-3β (GSK3β), that decreases its affinity for axonal microtubules and leads to its cytosolic accumulation. As previously mentioned in this review, P-tau is highly susceptible to aggregation, forming insoluble oligomers and NFTs; moreover, tau pathology in AD is suggested to be dependent on Aβ pathology, being correlated with cognitive impairment and clinical progression [5]. The propagation of tau aggregates is also an essential event in AD and is proposed to occur between specific and interconnected brain regions, that are typically affected in this pathology, namely the hippocampus and entorhinal cortex.

In this regard, previous studies revealed that ApoE4 promotes the accumulation of P-tau especially in CA3 hippocampal neurons of ApoE4-TR mice, which was accompanied by cognitive deficits [31], or accelerate tau pathology spreading particularly when secreted by neurons [50]. The mechanisms underlying these effects are unclear, but one hypothesis is related to ApoE4 binding to tau. To support this mechanism, ApoE4 has been demonstrated to interact directly with NFTs in AD brains, and it has been observed that this ApoE isoform potentiates the aggregation of P-tau into NFT-like inclusions in transfected primary neurons [51]. Another possible explanation for the increased tau pathology associated to ApoE4 is its effect on ApoE receptors. On the one hand, LRP1 also mediates tau uptake by neurons, being involved in tau spreading. Although the exact mechanisms are not understood, it has been observed that ApoE in general reduces LRP1-mediated tau uptake by neurons and propagation [52]. However, as previously described, ApoE4 in particular is associated to decreased total levels of ApoE [34], which is possibly related to the potentiation of tau pathology throw enhanced LRP-1 mediated tau propagation [52]. Moreover, ApoE4 activates LRP1 and subsequent intracellular signaling pathways, namely Wnt3/β-cathenin/GSK3β pathway, that ultimately potentiates GSK3β-mediated tau hyperphosphorylation [53,54], which may help to understand why ApoE4 is associated with an increased formation of NFTs formation and a greater risk to develop tau-mediated neurodegeneration.

Independent from glial ApoE4, neuronal ApoE4 was proven to be associated to increased neuronal vulnerability to cytotoxicity, Ca^2+^ dysregulation and cell death, in iPSCs derived from AD patients carrying the *APOE* ε4 allele. These alterations were accompanied by increased P-tau neuronal release and accelerated tau spreading, which supports the hypothesis that ApoE4 accelerates tau pathology propagation, through a trans-synaptic prion-like mechanisms [50].

In summary, ApoE4 aggravates tau pathology and potentiates tau-mediated neurodegeneration [2], which is associated with behavioral alterations, synaptic deregulation and neurodegeneration, and favors AD progression.

### 4.4. Neuroinflammation and Glial Cells—ApoE4 Pathogenic Mechanisms

Another pathological mechanism of ApoE4 is its influence on glial cells morphology and function. Although ApoE is involved in protective anti-inflammatory responses [24], ApoE4 per se has a negative effect on this role of glial cells compared to other isoforms.

Besides impairing the phagocytic role against Aβ, as previously described, ApoE4 potentiates inflammatory cascades, modifies microglial phenotype towards a proinflammatory profile and promotes APP-mediated activation of microglia [17], which induces neuroinflammation. Different studies have showed an association between this ApoE isoform and increased levels of neurotoxic and inflammatory cytokines, such as tumor necrosis factor α (TNFα), Interleukin-6 (IL-6), Interleukin-1β (IL-1β) and nitric oxide released by microglia and astrocytes, in different models of AD [24]. ApoE4-expressing astrocytes also show morphological alterations, with fewer and shorter processes and elongated cell bodies [55], which underlies astrogliosis and may also justify the impaired phagocytic ability of these cells.

In the context of astrocytic dysfunction, it has been proposed that ApoE4 is associated to a decreased ability of astrocytes to maintain Ca^2+^ and K^+^ homeostasis [24] and to control oxidative stress [56]. Although it is not understood how exactly ApoE4 affects these roles, it has been shown that astrocytes expressing ApoE4 exhibit higher cytosolic levels of Ca^2+^, mainly through ER inositol 1,4,5-trisphosphate receptor (IP_3_R)-mediated release [57].

## 5. Mitochondrial Alterations in AD—Impact of ApoE4

In previous studies, ApoE4 has been also associated to impaired mitochondrial function and dynamics, enhancing AD pathogenesis. This organelle is crucial for brain function, providing energy from glucose and is a primary site of action for Aβ and tau proteins [22], contributing to AD progression from early stages. Thus, it is important to unveil the mechanisms underlying mitochondrial alterations in AD, aggravated by ApoE4, aiming to develop therapeutic strategies targeting these early events.

Mitochondrial respiration is fundamental for energy homeostasis in the CNS, which is then crucial for neuronal viability. To ensure the high cerebral energy demand, respiratory chain complexes in mitochondria must maintain their integrity and function. Brain mitochondria in AD patients have shown dysfunctional respiratory chain complexes and decreased ATP generation, which has been observed for instance in purified COX (cytochrome c oxidase) from AD brains [58] and in a double-transgenic APP-mutant mouse model of early AD (AβPP_SL_ mice, expressing both Swedish and London mutation in human *APP* gene) [59]. Besides that, PET studies have suggested a marked cerebral glucose hypometabolism in brain tissue from LOAD patients decades before the symptoms arise, which was correlated with the decreased neuronal expression of Cx I-V subunits of the mitochondrial electron transport chain, in total brain extracts and in samples from specific brain areas, including the hippocampus and entorhinal cortex [60]. Apart from these studies, the activity of mitochondrial respiratory complexes was also demonstrated to be decreased in platelet and lymphocyte mitochondria, evidencing significative changes in Cx IV (*or* COX) activity, that is not only decreased in cells derived from advanced AD patients, but also from MCI individuals [61]. Other authors have proposed a parallel evolution of COX activity and cognitive impairment, based on brain samples from patients with different types of dementia, including AD [62]. Mutations in mitochondrial DNA (mtDNA) may be involved in mitochondrial dysfunction in AD, since they are more common in the brains of AD patients than in those of age-matched controls [22]. As such, neuropathological features due to Aβ accumulation have also been related with a decrease in respiratory capacity and ATP production, and changes in mitochondrial morphology [63], which can underlie mitochondrial dysfunction. Moreover, mitochondrial depolarization was observed in neurons subjected to oligomeric Aβ_1–42_ direct exposure [63].

Mitochondrial respiratory capacity, ATP production, Ca^2+^ levels and ROS generation are strictly related to mitochondrial membrane potential (Δψ_m_) [64]. Δψ_m_ is created by the ‘flux’ of protons from the mitochondrial matrix to the intermembrane space through the intrinsic respiratory chain complexes I, III and IV, alongside with the flux of electrons from reduced coenzymes, nicotinamide adenine dinucleotide, reduced form (NADH) and flavin adenine dinucleotide, reduced form (FADH_2_), generated by the Krebs cycle, up to O_2_ [65]. Mitochondrial dysfunction occurring in AD is accompanied by an impairment in Δψm. A decrease in Δψm (or mitochondrial depolarization) has been observed in several in vitro and in vivo AD models [65,66], being then another pathological feature of AD.

In other studies, it has been observed that Aβ accumulates within mitochondria in brain tissues from AD patients, which was linked to the hypothesis that mitochondrial dysfunction could trigger Aβ pathology [67]. Concordantly, it was further observed that membrane depolarization, decreased ATP levels, decreased COX activity and increased ROS levels occur before extracellular Aβ deposition and aggravate with age. These data were obtained in mitochondria isolated from AβPP_SL_ transgenic mice [68] and evidence the importance of studying mitochondria as a therapeutical target in AD.

Mitochondrial dysfunction is a hallmark of cellular senescence, being identified in neurons, astrocytes and microglia in the CNS. Apart from its role in ATP production, mitochondria are highly relevant for Ca^2+^ homeostasis. Dysfunctional mitochondria lose the ability to regulate Ca^2+^ levels, promoting synaptic dysfunction and cell death. Excessive Ca^2+^ levels in mitochondria activate pro-apoptotic pathways [69] and affect mitochondrial metabolism, ATP production and neurotransmitters release at the synaptic level [70], potentiating neurodegeneration.

Mitochondrial dynamics in turn is highly related with mitochondrial function. Impairments in mitochondrial fusion and fission processes affect mitochondrial size, shape and number. Moreover, alterations in mitochondrial degradation (through mitophagy or macroautophagy) and biogenesis interfere with mitochondrial health and activity. Several models of AD have demonstrated altered mitochondrial dynamics, not only in neurons, but also in astrocytes [71].

Tau pathology in AD is also correlated with mitochondrial alterations. Tau hyperphosphorylation and aggregation have been related to the impairment of mitochondrial dynamics, namely axonal transport and bioenergetics. Through alterations in mitochondrial localization in neurons, imbalance in fusion and fission processes and in ATP synthesis, ROS production and mitochondrial depolarization, aberrant tau phosphorylation and aggregation promote neuronal and synaptic damage, leading to cognitive decline in AD [66].

Taking into consideration that ApoE4 potentiates pathological pathways related to AD molecular hallmarks and cellular senescence and these processes are, in turn, interrelated with mitochondrial dyshomeostasis, it is relevant to study whether this genetic risk factor for LOAD perturbs mitochondrial health.

### 5.1. ApoE4 and Mitochondrial Dysfunction

The ε4 allele of the *APOE* gene has been consistently studied in the field of human longevity [20]. ApoE4 was the first genetic variation to be associated with decreased longevity [72]. This association is possibly linked to the neurotoxic effects of ApoE4, namely its impact on mitochondrial homeostasis, but it may also be related to the general poor cellular response to stress agents, the increased risk to develop premature atherosclerosis, or cardiovascular and neurocognitive deterioration [4,20,32]. Of relevance, several studies have demonstrated that carrying the ε4 allele of the *APOE* gene enhances AD-related mitochondrial dysfunction (Figure 1), accelerating disease progression.

In vitro studies in N2a cells showed that ApoE4 fragments (aa 1–272) interact with mitochondria and cause mitochondrial dysfunction and neurotoxicity [29]. As mitochondria play several roles in the cell, mitochondrial dysfunction may be approached through many parameters.

In the context of respiratory alterations, N2a cells stably expressing ApoE4 showed reduced levels of mitochondrial respiratory complexes (Cx) I, IV and V, while cortical neurons from NSE-ApoE4 showed reduced levels of all subunits, relatively to respective ApoE3-expressing controls. These alterations in N2a cells matched the decrease in mitochondrial total respiratory capacity, through a lower OCR (oxygen consumption rate) after stimulation with CCCP (Carboxyl cyanide *m*-chlorophenyl hydrazone), a mitochondrial uncoupler, in Seahorse experiments, relatively to ApoE3 expression [22]. This study proves that mitochondrial respiratory complexes are affected by ApoE4 expression, which then affects their activity and overall mitochondrial function. Accordingly, in other studies performed in ApoE4-expressing N2a cells, the authors demonstrated a decrease in respiration rates and ATP production, showing a poorer ability to respond to high energy demands relative to ApoE3-expressing cells [73]. Likewise, a decrease in mitochondrial respiratory complex IV activity was observed in autopsy brains from young ApoE4 carriers [74]; in addition, a characteristic glucose hypometabolism was observed in cognitively normal ApoE4 carriers [22].

These studies demonstrate that alterations in mitochondrial metabolism may appear before the clinical onset of AD and ApoE4 may promote an earlier manifestation of the disease.

Apart from the respiratory capacity, mitochondria play a critical role in Ca^2+^ buffering and the regulation of its intracellular flux. The maintenance of mitochondrial Ca^2+^ levels under a controlled range is crucial, since reduced levels of this ion may compromise mitochondrial bioenergetics and ATP production, and excessive mitochondrial Ca^2+^ levels induce mitochondrial permeability transition pore (mPTP) opening [70] and pro-apoptotic pathways [69]. In this perspective, an association between ApoE4 expression and mitochondrial Ca^2+^ overload was observed in in vitro models, such as Neuro-2 A cells [28,73], stem-cell-derived forebrain excitatory neurons [50] and rat hippocampal neurons [30]. In the last study, ApoE4 was demonstrated to cause an increase in basal mitochondrial Ca^2+^ levels and also an exacerbated neuronal Ca^2+^ entry in response to *N*-methyl-D-aspartate receptor (NMDAR) activation, which was correlated with neurotoxicity [30].

The association between ApoE4 and AD risk has been also linked to the function of mitochondria-ER-associated membrane (MAM), whose tethering facilitates the transport of Ca^2+^ from the ER to mitochondria and the exchange of lipids, being also a molecular platform essential for NLRP3 (NOD-like receptor protein 3, or nucleotide-binding oligomerization domain, leucine-rich repeat and pyrin domain-containing protein 3) inflammasome formation, among other potential cellular roles [74]. Some tethering proteins and Ca^2+^ channels are expressed in MAMs, being very important to regulate ionic flux between ER and mitochondria, and therefore for mitochondrial function and neuronal homeostasis. In this context, it has been demonstrated that MAM are overly activated and both organelles communicate more tightly in cells treated with ApoE4-containing astrocyte-conditioned medium, compared to ApoE3 [19]. Other studies have also shown an overexpression of some MAM proteins in N2a cells expressing ApoE4, including voltage-dependent anion-selective channel 1 (VDAC1) and inositol 1,4,5-trisphosphate receptor (IP_3_R), in agreement with a higher and faster Ca^2+^ flux from the ER to mitochondria, comparing to ApoE3 expression [73].

Another fundamental mitochondrial function is maintaining redox homeostasis through a balanced production of ROS and cellular antioxidant activity. Mitochondrial dysfunction is usually linked to excessive ROS production evolving to oxidative stress, and largely implicated in neurodegeneration. Also in this context, ApoE4 (272–299 aa fragment) was demonstrated to increase ROS levels in N2a cell line [28,73] (Figure 1), which was shown to correlate with increased ER stress and enhanced MAM formation by favoring mitochondria-ER direct interaction [28].

In terms of Δψ_m_ ApoE4 largest fragment (1–272 aa) was shown to directly interact with mitochondria, which leads to a decrease in Δψ_m_ and mitochondrial fragmentation [26,29,73]. However, this parameter is not completely understood, since other authors demonstrated that, under some transfection conditions, ApoE4 has no effect on Δψ_m_ [75].

### 5.2. ApoE4 and Impaired Mitochondrial Dynamics and Morphology

As mentioned above, mitochondrial function and dynamics are strictly related. Δψ_m_, for example, has been proven to control mitochondrial dynamics, which gives evidence to the importance of this association. Mitochondria undergo processes such as fusion, fission, mitophagy and transport along neurons and other cell types, being the balance of all these processes crucial for mitochondrial respiration, ATP production, oxidative homeostasis and neuroprotection [63].

Under cellular stress conditions, particularly in AD, changes in mitochondrial morphology is shifted towards immoderately enhanced fission, through changes in the expression of mitochondrial fission proteins, namely dynamin-like protein 1 (Drp1) and mitochondrial fission protein-1 (Fis1), and fusion proteins, namely mitofusin 1/2 (MFN1/2) and optic atrophy 1 protein (OPA1) [66,76]. This impairment has been observed in several in vitro models of AD, including APP/PS1 primary neurons, showing a tight relationship with hyperphosphorylated tau protein and microtubules disassembly [66].

Although the effect of ApoE4 on mitochondrial dynamics has not been studied in much detail, this protein has been associated to abnormally increased fusion, and decreased mitophagy and fission in the hippocampus of ApoE4-transgenic mice [18], and decreased fission and parkin-mediated mitophagy in APOE4 astrocytes [71]. However, other studies demonstrated an association between ApoE4 and fission activation and fusion inhibition [28], which may be related to ApoE4 fragmentation, namely the type of fragments that are generated [4]. Although it seems not to be consensual among the authors, these findings suggest that an imbalance in mitochondrial dynamics rather than a defined effect may underlie ApoE4-mediated mitochondrial dysfunction.

As previously mentioned, mitochondrial dynamics determines organelle size and morphology. Fragmentation of the mitochondrial network facilitates mitophagy, allowing the selective clearance of defective mitochondria [77] and consequently preventing excessive ROS production and Ca^2+^ dyshomeostasis. However, as mitochondrial fission is exacerbated in AD, mitochondria present an excessive level of fragmentation, being less elongated and more circular, and the loss of mitochondrial network integrity. Together with the downregulation of mitophagy, these alterations lead to the accumulation of dysfunctional mitochondria, that are unable to supply the energy neuronal demands, produce excessive ROS and Ca^2+^ levels, induce synaptic dysfunction and promotes Aβ and tau accumulation. These alterations will in turn contribute to AD progression and cognitive decline [2].

## 6. Transcriptional Regulation by ApoE4 

Another mechanism of action of ApoE4 that has been recently suggested is its translocation to the nucleus and potential role as a transcription factor [43].

It is not known how exactly ApoE4 enters the nucleus, but it has been shown that ApoE4, in its mature or fragmented form, especially the N-terminal, has a high affinity for DNA, linking to the promoter region of several genes. Therefore, this protein can act as a transcriptional regulator of approximately 1700 genes [43], not only in neurons, but also in glial cells. This transcriptional deregulation triggers processes such as synaptic dysfunction, inflammation and neuronal death [25], which contribute to the development of AD. ApoE4 was reported to directly bind to the promoter region of *SIRT1* gene, [37], decreasing its expression and enzymatic activity. Sirt1 is involved in normal aging and has neuroprotective effects, such as promoting autophagy [4,43], which may explain why ApoE4 impacts on its expression may be linked to neurotoxic effects.

ApoE4 also directly interacts with the promoter region of MAPK activating death domain (*MADD*), activity dependent neuroprotector homeobox (*ADNP*)*,* COMM domain containing 6 (*COMMD6*) and protein phosphatase 2, regulatory subunit B’, epsilon isoform (*PPP2R5E*) genes, decreasing their expression. MADD and COMMD6 are transcription factors that regulate gene expression by binding to the nuclear factor-κB (NFkB) complex, which is involved in several cellular processes, namely anti-apoptotic mechanisms. *ADNP*, in turn, acts as an anti-apoptotic and anti-inflammatory gene, while PPP2R5E forms a protein complex with protein phosphatase 2A (PP2A). Briefly, the referred alterations in gene expression associated to ApoE4 potentiate apoptotic and inflammatory pathways [37,43] (Figure 1).

Another protein which expression is downregulated by the direct transcriptional regulation of ApoE4 is PP2A, which is related to a decrease in tau dephosphorylation [43], thus contributing to tau hyperphosphorylation, an important hallmark of AD. It has been also suggested that ApoE4 inhibits Drp1 transcription and expression, which may be linked to the disruption of mitochondrial fission [71].

Apart from its direct transcriptional role, ApoE4 has also an indirect impact on gene expression, by acting as a signaling molecule and intracellular messenger. For example, it signals for histone deacetylase (HDAC) nuclear translocation, decreasing BDNF expression, which is crucial for synaptic plasticity. By activating a MAPK cascade, ApoE4 also promotes APP expression and Aβ production, proposing a clear impact on AD progression [43].

In prefrontal cortical tissues from post-mortem human brains, various molecular processes were found to be altered by ApoE4, not only in neurons, but also in glial cells, namely in oligodendrocytes. Amongst them, the authors identified an upregulation of inflammatory and immune-related pathways, a downregulation of processes related to synaptic plasticity, channel activity and excitatory post-synaptic potential, and significant changes in APP- and Aβ-related metabolic pathways [36]. Corroborating the role of ApoE in regulating brain cholesterol metabolism, these authors demonstrated that ApoE4 is associated to altered cholesterol and other lipids metabolic pathways in distinct cell types, through modified transcriptomics [36].

ApoE4 also modulates the action of another transcription factor, nuclear respiratory factor 1 (NRF1), possibly because the single nucleotide polymorphism generates a binding motif for NRF1 that underlies the ability of ApoE4 to act as a transcription factor itself [78]. By affecting NRF1 transcriptional role, ApoE4 then influences the expression of its target genes, potentially involved in mitochondrial dynamics and Ca^2+^ handling [37]. Other studies demonstrated that ApoE4 is associated to a deregulation in mRNA levels of genes encoding for subunits of mitochondrial respiratory complexes (Figure 1), encoded by nuclear and mitochondrial DNA [22], as previously explored in this review. Supporting the impact of ApoE4-induced transcriptional changes on organelle function, a recent study showed the upregulation of molecular pathways related to cellular and ER stress, namely in chaperone-mediated protein folding and unfolded protein response, in distinct cell types of human post-mortem brain tissue [36].

In summary, the relationship between ApoE4, AD and mitochondrial function and dynamics is straightforwardly connected to ApoE4 role as a transcription factor. All previously suggested mechanisms, although not completely understood, seem to be related to mitochondrial dysfunction, neurodegeneration and AD progression. As mitochondrial alterations are an early feature of AD and ApoE4 is the major genetic risk factor for the disease, the study of its mechanisms may be crucial to determine therapeutical targets.

## 7. ApoE-Targeted Therapeutic Strategies

Some therapeutic and protective strategies that have been suggested include the increase of ApoE2 expression in *APOE* ε4 carriers, through vector administration or liposomal delivery systems [21]. Adeno-associated viruses (AAV) carrying the ApoE2 genetic information were used in animal models to overexpress this isoform of ApoE in the brain, through intracerebral or intracisternal viral injection, e.g., [21]. Using this gene delivery approach, it was demonstrated that overexpression of human *APOE* ε2 allele in APP/PS1 mice was able to attenuate brain Aβ oligomerization and plaque deposition, as well as peri-plaque synapse loss and neurite dystrophy, in contrast to the injection of ApoE4-encoding AAVs [79]. A clinical trial using this approach is currently in phase I (NCT03634007: ‘Gene Therapy for APOE4 Homozygote of Alzheimer’s Disease’), aiming to assess the safety and toxicity of intracisternal administration of AAVrh. 10 hAPOE2 in *APOE4* homozygotes with AD, aged 50 or older, and potentially increase CSF ApoE2 levels. Liposomes were also used for this purpose, and the ability to modify the surface of these molecules may enable targeting specific cells in the CNS, enhancing the efficiency of gene delivery. A previous study showed that intravenous administration of ApoE2-encoding plasmid DNA, using surface-functionalized liposomal delivery system targeting Glut-1 (glucose transporter-1) and containing a cell-penetrating peptide, was highly efficient in overexpressing this isoform of ApoE in the mouse brain; this strategy also showed high transfection rates in cultured neurons below the in vitro model of blood brain barrier [80]. These data suggest a safe and effective AD gene therapy using surface-modified liposomes for brain-targeted delivery of *APOE2* gene.

Another therapeutic strategy consists in using antisense oligonucleotides (ASO) to downregulate the expression of ApoE4 [21]. These molecules can target specific molecules and genetic alterations based on sequence complementarity, thus allowing to modulate gene expression [81]. A previous study in APP/PS1 mice showed that administration of ASOs lowered ApoE levels in *APOE* ε4 homozygous mice, which prevented Aβ plaque formation or related toxicity in an age-dependent way [82]. The administration of ApoE-targeted ASO in an ApoE4-expressing mouse model of tauopathy, which is defined by the tau pathology common to AD, was also demonstrated to protect against tau-associated neurodegeneration and other neuropathological features of this disease, namely synaptic disruption and neuroinflammation [83].

Lowering the levels of ApoE4 protein through immunotherapy, using monoclonal antibodies against this isoform, is another approach that has shown promising experimental results. These antibodies are proposed to specifically target brain ApoE4, diminishing ApoE4 toxicity [84]. Direct intracerebroventricular injection of mAb 9D11 prevented ApoE4-mediated accumulation of Aβ in hippocampal neurons by inhibiting neprilysin, an Aβ-degrading enzyme. Thus, administration of anti-ApoE4 antibodies attenuates Aβ pathology, neurodegeneration and cognitive impairments in ApoE4-expressing mouse models of AD [84]. These studies suggest that increasing the levels of ApoE2 or lowering the levels of ApoE4 would be protective against AD-related pathogenic mechanisms, including those related to the main histopathological hallmarks of the disease.

Another line of reasoning, based on the hypothesis that neuronal ApoE4 fragmentation is a key event for ApoE4-induced neurotoxicity, suggests that preventing its proteolytical cleavage into fragments could be therapeutically effective in AD. In this regard, small molecules that disrupt the interaction between the N- and C-terminal domains of ApoE4 protein, also called “structure correctors”, have been tested in such a way that ApoE4 becomes less prone to fragmentation [85]. In this respect, ApoE4-expressing N2a cells pharmacologically treated with GIND25, a small molecule that is able to disrupt ApoE4 domain interaction, reverted some AD-related mitochondrial alterations, namely by restoring Cx IV levels [22]. By blocking ApoE4 domain interaction, GIND25 was then able to modify the protein tertiary structure into an ApoE3-like conformation, which is proposed to underly the ablation of its neurotoxic effects [85]. Additionally, treating ApoE4-expressing iPSC-derived neurons with PH002, another structure corrector, was able to ameliorate the toxic effects induced by this isoform of ApoE, by reducing ApoE4 detrimental effect on neurite outgrowth in Neuro-2a cells and dendritic spine development in primary neurons [85], increasing the number of GABAergic neurons and reducing P-tau, Aβ_40_ and Aβ_42_ levels [86]. The potentially therapeutic effects induced by PH002 were accompanied by a significant decrease in ApoE4 fragment levels [86], which corroborates the hypothesis that inhibiting ApoE4 domain interaction may be a key event in preventing its proteolytical cleavage and subsequent neurotoxic effects.

In the context of targeting ApoE4 fragmentation as a therapeutical strategy in AD, the inhibition of specific proteases is proposed to be a promising approach; however, a more detailed study about the enzymes involved in neuronal ApoE4 cleavage is still necessary [11,21,45]. Nevertheless, as previously mentioned in this review, some authors identified collagenase, MMP-9, neuron-specific α-chymotrypsin-like serine protease, cathepsin D and HtrA1 as some of the proteases that are able to cleave ApoE4 into neurotoxic fragments. Thus, finding molecules that can block its enzymatic activity would be relevant to prevent their deleterious effects.

To genetically convert ApoE4 into ApoE3 or ApoE2 proteins would be also hypothetically effective in reducing ApoE fragments since their structural properties make these proteins less susceptible to proteolytical cleavage; this would also protect against AD-related phenotypical alterations that have been extensively attributed to ApoE4 [17]. CRISPR/Cas9 technology, which allow precise genetic modifications, was used to convert *APOE* ε4 into *APOE* ε3 in neuronal, astrocytic and microglial cells types from iPSCs derived from LOAD patients, which led to the attenuation of some AD-related pathologies, namely Aβ glial uptake [39]. Despite its promising results in vitro, genome editing has not yet revealed effective results in animal models [45] and its translation to clinics might be challenging.

As explained in this review, ApoE plays an important role in mediating lipid transport and, consequently, facilitating Aβ clearance. Thus, taking into consideration the contribution of Aβ accumulation and aggregation to neurodegeneration and AD progression, lipid metabolism is another hypothetically promising therapeutical target. Comparing to the other ApoE isoforms, ApoE4 is hypolipidated, which prevents its binding to soluble and fibrillar forms of Aβ and enhances its accumulation. Thus, using ABCA1 agonists allows ApoE4 lipidation and was shown to reduce ApoE4-mediated Aβ42 accumulation and tau hyperphosphorylation in hippocampal neurons, as well as rescue synaptic impairments and cognitive deficits in ApoE4-targeted replacement mice [87]. ABCA1 agonists also rescued impaired Aβ degradation in ApoE4 cells and further decreased ApoE and ABCA1 aggregation in the hippocampus of ApoE4-targeted replacement mice [88].

Another recently proposed strategy to attenuate ApoE4-mediated impairments in brain lipid metabolism was to increase cholesterol transport. Cyclodextrin, an inhibitor of cholesterol biosynthesis, was able to reduce cholesterol accumulation in oligodendrocytes, rescuing their function and promoting myelination in vitro and in vivo. Pharmacological treatment of ApoE4-TR mice with this molecule was also able to improve learning and memory function [36].

Several other therapeutical approaches have been consistently proposed, reinforcing the relevance of using the up-to-date knowledge to develop new strategies towards a better comprehension of ApoE4 pathogenic pathways and intracellular mechanisms, in order to identify putative therapeutical targets and reduce the vulnerability to develop AD, ApoE4 neurotoxic effects and subsequent cognitive decline. Summary of ApoE-related therapeutic strategies and respective experimental-based approaches is displayed in Table 1.

## Figures and Tables

**Figure 1 ijms-24-00778-f001:**
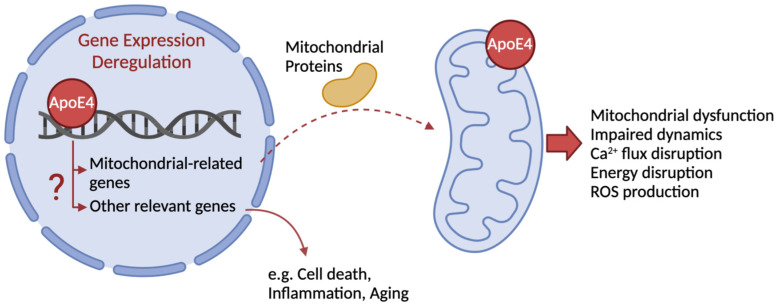
Schematic representation of ApoE4 subcellular localization and potential effects on transcriptional regulation and mitochondrial processes. Previous studies have suggested that ApoE4 acts as a transcription factor, modulating the expression of mitochondrial-related genes and other genes linked to neurodegeneration. ApoE4 may also directly interact with mitochondria, affecting mitochondrial function and dynamics through modified metabolism and fusion/fission imbalance, respectively.

**Table 1 ijms-24-00778-t001:** Summary of ApoE-related therapeutic strategies and respective experimental-based approaches.

Therapeutic Strategy	Experimental-Based Approach	Reference(s)
ApoE2 brain delivery	AAVs encoding ApoE2 expression; Surface-functionalized liposomes containing ApoE2 plasmid	Hudry et al. [79] Arora et al. [80]
Downregulation of ApoE4 expression	ASOs targeting ApoE4	Huynh et al. [82]; Litvinchuk et al. [83]
Decrease in ApoE4 levels	Immunotherapy using anti-ApoE4 monoclonal antibodies	Luz et al. [84]
Prevention of ApoE4 proteolytical cleavage into neurotoxic fragments	Small molecules inhibiting ApoE4 domain interaction (“structure correctors”)	Brodbeck et al. [85]; Wang et al. [86]
Genetic conversion of *ApoE ε4* allele into *ε3 or ε2*	CRISPR-Cas9 technology	Lin et al. [39]
Increase in ApoE4 lipidation to promote LRP1-mediated Aβ clearance	ABCA1 agonists	Boehm-Cagan et al. [87]; Rawat et al. [88]
Increase in cholesterol transport and myelination	Cholesterol biosynthesis inhibitors	Blanchard et al. [36]

## Data Availability

Data sharing is not applicable to this article.

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
