# Peer review of "Apoe4 and Alzheimer’s Disease Pathogenesis—Mitochondrial Deregulation and Targeted Therapeutic Strategies"

_ijms, 2023, doi:10.3390/ijms24010778_

Round 1
Reviewer 1 Report
This review by Mariana Pires, Ana Cristina Rego on ApoE4 and AD disease pathogenesis and focusing on mitochondrial and targeted therapeutic strategies.
I appreciate the authors for the well-constructed review, which I believe is useful to the field. I have only one minor concern. I suggest authors discuss extensively the recent publication (https://doi.org/10.1038/s41586-022-05439-w) which highlights the role of APOE4 in impaired myelination.
Author Response
In accordance with the suggestion made by the reviewer, the article by Blanchard and co-authors (Nature, 2022) is now described in the following sections:
a) “4. Physiological and pathological roles of ApoE in the central nervous system” – page 7;
b) “6. ApoE4 and AD development” – page 10;
c) “10. Transcriptional regulation by ApoE4” – pages 19 and 20; and
d) “11. ApoE-targeted therapeutic strategies” – page 22 (last paragraph).
In this version of the Ms, we numbered the different sections and sub-sections of the review.

Reviewer 2 Report
In this manuscript by Mariana Pires and Cristina Rego, authors aim to detail the impact of genetic risk factor for late on set AD (LOAD) and mitochondrial dyshomeostasis, and discussed ApoE4-targeted therapeutic strategies based on ApoE4 potential mechanism of action.
Over all the manuscript is well written and presented all the latest literature concerning the ApoE4 and AD pathogenesis in mitochondrial deregulation.
Specific comments:
1) In introduction, page 1 there is a statement saying "AD is the main cause of disability in elderly" (REF?, arthritis should be the cause), this should be elaborated and properly justified.
2) Sections 9-11 should be consolidated and be better if they are same sections with different sub sections, the overall message coming from each section look same but they are elongated.
3) Section 12 appeared twice in the manuscript
4) The pictorial representations in the manuscript needed to be significantly improved and they are not justifying the text. Refer the paper below with similar outline and modify the images
"Kodavati M, Wang H, Hegde ML. Altered Mitochondrial Dynamics in Motor Neuron Disease: An Emerging Perspective. Cells. 2020 Apr 24;9(4):1065. doi: 10.3390/cells9041065. PMID: 32344665; PMCID: PMC7226538."
5) The therapeutic strategies proposed are too vaguely described, should be re-written and a table describing each strategy will help
Author Response
Specific comments:
1) In introduction, page 1 there is a statement saying "AD is the main cause of disability in elderly" (REF?, arthritis should be the cause), this should be elaborated and properly justified.
R: In Introduction (Ms page 3), the paragraph referring to AD description was improved to include impairments in language, social behavior and visuospatial function. Furthermore, and in accordance with the reviewer’s suggestion, we altered the sentence when referring to AD as a cause of disability in aging: “AD is considered one of the main causes of disability in the elderly population [2,5]. Indeed, (…)”.
2) Sections 9-11 should be consolidated and be better if they are same sections with different sub sections, the overall message coming from each section look same but they are elongated.
R: In this version of the Ms, we numbered the different sections of the review, and included the description of ApoE4 role on mitochondrial dysfunction (section 9.1) and altered dynamics/morphology (section 9.2) as part of section 9, entitled “Mitochondrial alterations in AD – impact of ApoE4”. To avoid overlapping information, the paragraph referring to ApoE4 in section 9 was moved to section 9.1 (page 15).
3) Section 12 appeared twice in the manuscript
R: The authors could not find this error in the text. Numbering of each section in the current version of the Ms may help to better visualise them.
4) The pictorial representations in the manuscript needed to be significantly improved and they are not justifying the text. Refer the paper below with similar outline and modify the images
"Kodavati M, Wang H, Hegde ML. Altered Mitochondrial Dynamics in Motor Neuron Disease: An Emerging Perspective. Cells. 2020 Apr 24;9(4):1065. doi: 10.3390/cells9041065. PMID: 32344665; PMCID: PMC7226538."
R: Although Figure 1 (and potentially other figures) could include the observations described along the Ms, the authors decided to make it simpler and focus on one of the main themes of the review – the potential impact of ApoE4 on mitochondrial function. Based on this, additional information could be depicted based on interactions of ApoE4 with fission and fusion proteins; however, this has been poorly studied so far.
Figure 1 is now more often indicated along the Ms, to better link the text to the image representation(s).
5) The therapeutic strategies proposed are too vaguely described, should be re-written and a table describing each strategy will help.
R: In accordance with the suggestion made by the reviewer, we detailed the description of the different ApoE-related therapeutic strategies (section 11, pages 20-22) and included Table 1 to resume the therapies and the respective experimental-based approaches.

Round 2
Reviewer 2 Report
Authors response are satisfying